# Design and Fabrication of Nanofibrous Dura Mater with Antifibrosis and Neuroprotection Effects on SH-SY5Y Cells

**DOI:** 10.3390/polym14091882

**Published:** 2022-05-05

**Authors:** Zhiyuan Zhao, Tong Wu, Yu Cui, Rui Zhao, Qi Wan, Rui Xu

**Affiliations:** 1Department of Interventional Radiology, The Affiliated Hospital of Qingdao University, Jiangsu Road 16, Qingdao 266000, China; qduzzy@126.com (Z.Z.); zhrui97@163.com (R.Z.); 2Institute of Neuroregeneration and Neurorehabilitation, Qingdao University, Qingdao 266071, China; twu@qdu.edu.cn (T.W.); cuiyu1216@126.com (Y.C.); 3Qingdao Medical College, Qingdao University, Qingdao 266071, China

**Keywords:** nanofibrous dura mater, antifibrosis, neuroprotection, PLGA, tetramethylpyrazine

## Abstract

The development and treatment of some diseases, such as large-area cerebral infarction, cerebral hemorrhage, brain tumor, and craniocerebral trauma, which may involve the injury of the dura mater, elicit the need to repair this membrane by dural grafts. However, common dural grafts tend to result in dural adhesions and scar tissue and have no further neuroprotective effects. In order to reduce or avoid the complications of dural repair, we used PLGA, tetramethylpyrazine, and chitosan as raw materials to prepare a nanofibrous dura mater (NDM) with excellent biocompatibility and adequate mechanical characteristics, which can play a neuroprotective role and have an antifibrotic effect. We fabricated PLGA NDM by electrospinning, and then chitosan was grafted on the nanofibrous dura mater by the EDC-NHS cross-linking method to obtain PLGA/CS NDM. Then, we also prepared PLGA/TMP/CS NDM by coaxial electrospinning. Our study shows that the PLGA/TMP/CS NDM can inhibit the excessive proliferation of fibroblasts, as well as provide a sustained protective effect on the SH-SY5Y cells treated with oxygen–glucose deprivation/reperfusion (OGD/R). In conclusion, our study may provide a new alternative to dural grafts in undesirable cases of dural injuries.

## 1. Introduction

The dura mater surrounds the brain and retains the cerebrospinal fluid [1]. The dura mater may be damaged in the development and treatment of neurosurgical diseases, such as large-area cerebral infarction, cerebral hemorrhage, brain tumor, and craniocerebral trauma, which need to be repaired in time [2]. For dura mater that is difficult to complete in a one-stage repair process, it can be repaired with dural grafts through duroplasty [3]. Autologous dura grafts obtained from the periosteum and fascia have no immune rejection, but their clinical application is greatly limited due to insufficient quantities, difficult sampling, postoperative pain, and other shortcomings [4,5]. Dural grafts have several important features: supporting tissue regeneration, avoiding an immune inflammatory response, good watertight confinement, anti-adhesion, inhibiting scar tissue formation, biodegradability, and releasing therapeutic drugs to promote recovery [6,7].

Many synthetic polymers, such as polylactic acid (PLA), polyglycolic acid (PGA), and their copolymer, polylactic-co-glycolic acid (PLGA), have been widely used in biomedical engineering due to their good biocompatibility, non-toxicity, film-forming properties, and biodegradability [8,9]. In addition, the use of PLGA as a carrier for sustained drug release is also a hot topic of current research due to the controlled degradation rate of PLGA [10,11,12]. However, PLGA also has defects as a graft. The degradation products of PLGA are lactic acid and hydroxyacetic acid, which are by-products of human metabolism [13]. The accumulation of acidic degradation products can decrease the local pH value, trigger inflammatory reactions, and affect the rate of polymer degradation [14].

As a natural polysaccharide with good biocompatibility, degradability, antibacterial properties, and mechanical strength, chitosan (CS) is a kind of good implant material [15]. Previous studies have shown that CS can inhibit the proliferation of fibroblasts and suppress type I and III procollagen production, and it has anti-fibrosis as well as anti-adhesive effects [16,17,18,19]. In addition, chitooligosaccharides (COS), the degradation products of CS, have been proven to promote nerve regeneration [20,21]. CS is an alkaline polysaccharide among natural polysaccharides that can chemically bind to other substances through some primary amino groups carried by the CS main chain. It has been reported that CS can buffer the acidity produced from the degradation of PLLA [22,23].

2,3,5,6-tetramethylpyrazine (TMP) is an active compound extracted from the herb Chuanxiong Ligusticum, which exhibits neuroprotective effects [24,25]. TMP can protect neurons by scavenging oxygen free radicals, protecting mitochondrial function, inhibiting calcium inward flow and glutamate release, as well as attenuating ischemia-induced neuronal death by regulating the expression of bcl-2 and bax proteins [26,27,28,29,30]. In addition, TMP affects neurogenesis, for example, it can promote the differentiation of neural stem cells into neurons [31,32]. At present, TMP has been widely used in the clinical treatment and basic research of cardiovascular and cerebrovascular diseases. However, due to its poor water solubility, short half-life, and low concentration of distribution at the injury site, it requires high doses and multiple administrations to maintain therapeutic concentrations, which is a drawback for clinical treatment [33,34,35].

To overcome the above drawbacks and limitations, in this study, we designed and fabricated PLGA/TMP nanofibrous dura mater (NDM) with a coaxial electrospinning technique, and then generated PLGA/TMP/CS NDM by chemically binding a CS coating to the surface of the NDM. Our study found that the PLGA/TMP/CS NDM could inhibit the excessive proliferation of fibroblasts in vitro, thus exerting anti-adhesive effects and inhibiting the formation of scar tissue. Through the degradation of the PLGA and CS, TMP was released into the cell matrix, which could promote the survival of OGD/R-treated SH-SY5Y cells, as well as facilitate the regeneration of SH-SY5Y cells, and finally, exert a sustained neuroprotective effect.

## 2. Materials and Methods

### 2.1. Materials

PLGA (50:50, Mw: 60,000–80,000) was purchased from Match Biomaterials (Shenzhen, China), TMP and MES Buffered Solution from Aladdin (Shanghai, China), ethylcarbodiimide hydrochloride (EDC), *N*-hydroxysuccinimide (NHS), acetic acid, hexafluoroisopropanol (HFIP), and *N,N*-dimethylformamide (DMF) from Macklin (Shanghai, China), and CS from Sigma-Aldrich (America). Fibroblasts were a gift from Ziyi Zhou from the Medical Cosmetology Center at the Affiliated Hospital of Qingdao University. Human neuroblastoma SH-SY5Y cells were provided by the Institute of Neuroregeneration and Neurorehabilitation, Qingdao University. Dulbecco’s modified eagle medium (DMEM) and antibiotic-antimycotic were purchased from Solarbio (Beijing, China). Fetal bovine serum (FBS) was purchased from Pan (Adenbach, Germany). Phalloidin-iFluor 488 and DAPI staining solution were acquired from Abcam (Shanghai, China). The following antibodies from ABclonal (Wuhan, China) were used: Ki67 Rabbit pAb and GAP43 Rabbit pAb. Goat Anti-rabbit IgG H&L/Alexa Fluor 555 as a secondary antibody was from Bioss (Beijing, China). A Cell Counting Kit-8 (CCK8) was supplied from Targetmol (Boston, MA, USA). 

### 2.2. Electrospinning

PLGA/TMP/CS NDM was fabricated by the electrospinning technique. Briefly, PLGA was dissolved in HFIP via magnetic stirring at room temperature for 4 h to form an electrospinning solution with a concentration of 20 wt.% [36]. The solution flowed out from the syringe at a rate of 2 mL/h. PLGA NDM was fabricated by electrospinning for 1 h with +12 kV, and the acceptance distance was 15 cm. The PLGA NDM was immersed in EDC/NHS solution (0.96 g of EDC and 0.14 g of NHS in 50 mL of MES buffer) at 4 °C for 12 h. The CS solution (3 wt.%) at an equal volume to the EDC/NHS solution was added to ensure an excess of CS, and the mixed PLGA-CS system was kept for 24 h at room temperature until the cross-linking between the PLGA and CS components (via the coupling reaction between the NHS-ester groups belonging to PLGA and some primary amine groups of CS) was accomplished [12]. Then, the cross-linked product was rinsed repeatedly with deionized water and dried to obtain PLGA/CS NDM. For coaxial electrospinning, the shell solution consisted of 20% PLGA in HFIP, and the core of the fibers was 10 mg/mL TMP dissolved in ethanol. The prepared solutions were delivered to the outer and inner coaxial needle at 2.0 and 0.1 mL/h feeding ratios, respectively, with a programmable syringe pump. The applied voltage was 12 kV and the acceptance distance is 15 cm. The fibers were collected in a layer-by-layer manner during a 4 h period to obtain PLGA/TMP/CS NDM [36,37].

### 2.3. NDM Characterization

The morphology of the nanofiber was observed using scanning electron microscopy (SEM, VEGA 3 SBH, TESCAN, Shanghai, China) and the nanofiber diameter was measured by applying Nano Measurer software. The chemical structure and composition of the fibers were characterized by Fourier infrared spectroscopy (Nicolet Is50, Thermo Electron Corporation, Waltham, MA, USA). All NDMs were prepared for the same size (5 cm × 1 cm, about 0.02 mm in thickness) to characterize their tensile mechanical property by employing an Electro-mechanical Universal Testing Instrument (CMT6103, Mechanical Testing & Simulation, Eden Prairie, MN, USA). Thermogravimetric analysis (TGA) of the NDMs was performed using a Thermal Gravimetric Analyzer (TASDT650, TA INSTRUMENTS, New Castle, DE, USA).

### 2.4. Encapsulation Efficiency (EE) 

The absorbance value of the TMP was detected using a full-function microplate detector (Synergy Neo2, Bio Tek, Vermont, USA) to determine the maximum UV absorption peak at a wavelength of 280 nm (Appendix A). The PLGA/TMP/CS NDM samples were immersed in DCM until the TMP was completely dissolved, and then its actual content was determined spectrophotometrically using a corresponding calibration curve. Comparatively, the theoretical content of the TMP was considered to be the overall TMP amount consumed during the electrospinning. The calculation of the EE (in %) of the TMP was performed as follows:EE = (actual content/theoretical content) × 100(1)

### 2.5. In Vitro TMP Release Profiles

The PLGA/TMP/CS NDM was immersed in 5 mL of PBS solution (pH = 7.4) as a release medium and placed in a thermostatic shaker for gentle shaking. For current measurements, 1 mL of the PBS solution was taken out at different times to spectrophotometrically determine the TMP release (using the corresponding calibration curve), and then 1 mL of fresh PBS solution was added to the release medium contained in the shaker. The cumulative TMP release (C, in %) was calculated according to the following equation: C = (m_1_ + m_2_ + …+ m_n_)/m_0_ × 100%(2)
where m_1_, m_2_, and m_n_ are the weights determined at the times t_1_, t_2_, and t_n_, respectively, and m_0_ is the total weight of TMP to be released.

### 2.6. Extraction of NDM Immersion Solution

Multiple different NDMs with the same mass were sterilized by ethanol fumigation for 3 h and UV irradiation for half an hour. The NDMs were immersed in 8 mL of fresh DMEM complete medium (DMEM with 10% FBS) under aseptic conditions at 37 °C, while a separate fresh DMEM complete medium was used as a control. The NDMs were removed after 1, 4, 7, and 14 days of immersion from the medium, and the remaining medium was the NDM immersion solution.

### 2.7. Cell Culture

Fibroblasts and SH-SY5Y cells were cultured using DMEM complete medium containing 10% FBS at 37 °C and 5% CO_2_. Depending on the needs of the experiment, fibroblasts were seeded evenly on glass slides and different NDMs of a certain number, while SH-SY5Y cells were grown directly onto well plates and glass slides of a certain number. The culture medium was replaced every two days during the culture. Prior to cell seeding, all the glass slides and NDMs were sterilized by ethanol fumigation for 3 h and UV irradiation for 30 min.

### 2.8. OGD Challenge

The complete medium was replaced with deoxygenated glucose-free extracellular solution (in mM: 116 NaCl, 5.4 KCl, 0.8 MgSO_4_, 1.0 NaH_2_PO_4_, 1.8 CaCl_2_, and 26 NaHCO_3_) [38]. The cells were cultured in a dedicated chamber (Plas-Labs, Lansing, MI, USA) with 95% N_2_/5% CO_2_ at 37 °C for 3 h. Then, the cells were reperfused using fresh complete medium containing different immersion solutions according to the experimental requirements and transferred to normal conditions for culture.

### 2.9. Cell Viability

A cell viability assay for fibroblasts and SH-SY5Y cells was performed following an algorithm reported elsewhere [39]. The fibroblasts were seeded evenly at a density of 5000 cells/well on glass slides and different NDM amounts in 24-well plates for culture. CCK8 solution was added and incubated with the cells for 3 h at 1, 3, and 5 days after culture, respectively. The absorbance value at 450 nm was measured using a 96-well plate reader. The SH-SY5Y cells were seeded in 48-well plates at a density of 5000 cells/well, and after 24 h of culture, the cells were treated with OGD/R, after which CCK8 solution was added and incubated for 2 h. The absorbance value at 450 nm was measured using a 96-well plate reader. The absorbance values at 450 nm corresponded to the amounts of formazan dye that resulted under the action of cellular dehydrogenases exerted on the tetrazolium salt present in the initial CCK8 solution which, in turn, was proportional to the number of living cells.

### 2.10. Lactate Dehydrogenase (LDH) Release Assay

According to the manufacturer’s instructions (Beyotime, Shanghai, China), the supernatants from the OGD/R-treated cell culture medium were harvested, and the absorbance value at 490 nm was measured using a 96-well plate reader with the absorbance value at 600 nm as reference. The LDH release was calculated according to the manufacturer’s formula. The absorbance values at 490 nm were proportionally correlated with the LDH release.

### 2.11. Cell Morphology

The fibroblasts were seeded at a density of 5000 cells/well on glass slides, the PLGA NDM and PLGA/CS NDM in 24-well plates for culture. After 5 days of culture, the fibroblasts were stained with Phalloidin-iFluor 488 and DAPI to observe the morphology using fluorescence microscopy. For the SH-SY5Y cells, they were stained and observed in the same way after OGD/R treatment.

### 2.12. Immunofluorescence

Cells on the NDM and glass slides were fixed in 4% paraformaldehyde, then permeabilized with 0.5% Triton X-100 for 15 min and blocked in 5% FBS for 2 h. The primary antibodies against Ki-67 (1:200) and Gap43 (1:200) diluted in the blocked buffer were added to incubate with cells at 4 °C overnight. Then the cells were labeled by a secondary antibody and stained with DAPI. The samples were observed using fluorescence microscopy and analyzed using Image J software.

### 2.13. Statistical Analysis

The statistical analysis was performed using GraphPad Prism software. The results are presented in the form of the mean ± standard deviation. Statistical comparisons between groups were performed using one-way ANOVA. Values of * *p* < 0.05, ** *p* < 0.01, and *** *p* < 0.001 are considered statistically significant.

## 3. Results and Discussion

### 3.1. NDM Characterization

We fabricated PLGA NDM and PLGA/TMP NDM using an electrospinning device according to the previously described conditions and coated them with CS by EDC/NHS cross-linking to prepare PLGA/TMP/CS NDM. We observed the morphology of the NDM by SEM. A smooth surface structure was shown on the fibers of the PLGA NDM, which were arranged in a disordered manner and interwoven into a network. The mean diameter of the fibers was 646 ± 103 nm, which was relatively uniform (Figure 1A). The PLGA/TMP NDM was morphologically similar to the PLGA NDM, with 692 ± 97 nm in mean fiber diameter (Figure 1B). For the PLGA/CS NDM, shiny granular agglomerates could be seen on the fiber surfaces (Figure 1C). To determine whether these were made of CS, structural analysis on the molecular scale of the PLGA/CS NDM was performed via infrared spectroscopy (FTIR) (Figure 1D). In the PLGA spectrum, the peak at 1087 cm^−1^ was assigned to C-O stretching, that located at 1748 cm^−1^ to C=O stretching vibrations, and the peaks placed between approximately 2950 and 3000 cm^−1^ to C-H stretching vibrations mainly involving methyl (CH3) and methylene (CH2) groups [40]. Instead, the infrared spectrum of the CS displayed the two vibrations of amide I and amide II at 1646 and 1587 cm^−1^, respectively. These weak peaks ascribed to the vibrational mode of amide I and II are due to a high degree of N-deacetylation associated with the CS used. In addition, the band at 2870 cm^−1^ was attributed to C-H stretching involving the carbon atoms of the sugar rings. The broad band centered at ca. 3500 cm^−1^ corresponded to the N-H stretch vibrations overlapped by the O-H stretches of the OH groups [41,42]. On the other hand, the simultaneous presence of the bands at 3500, 1635, and 1536 cm^−1^ (for CS) and at 1748 cm^−1^ (for PLGA) on the PLGA/CS NDM spectrum confirms the coexistence of CS and PLGA in the same mixed system (PLGA/CS NDM). Moreover, the strengthening of the bands at 1635 and 1536 cm^−1^ on the IR spectrum of the PLGA/CS NDM is consistent with the increased number of amide cross-linkages newly formed during the grafting of CS onto PLGA via the EDC/NHS coupling reaction [12]. 

The tensile results of all NDMs are shown in Figure 2A as the stress–strain curves of the PLGA NDM and PLGA/CS NDM. The tensile strength of the PLGA NDM was 6.27 ± 0.96 MPa, while that of the PLGA/CS NDM is 8.71 ± 1.03 MPa. Correspondingly, the values of the maximum strain (at break) are ca. 216% for the PLGA NDM and 161% for the PLGA/CS NDM. It is obvious that the CS improved the mechanical stress of the PLGA NDM but had a negative effect on the flexibility. According to experimental data reported elsewhere [43], the tensile strength of the human dura mater is about 7 MPa, and the maximum strain is 11%. The result suggests that the PLGA/CS NDM is more suitable for a dural graft than the PLGA NDM.

The thermal stability of the NDMs was characterized using thermogravimetry (Figure 2B). The NDMs showed two stages of weight loss. From 30 to 200 °C, the PLGA NDM showed a gradual weight loss of 7.9 wt.%, while the PLGA/CS showed a 6.52% loss, and the PLGA/TMP NDM showed a 9.32% loss due to the desorption of solvent, adsorbed, and bound water. Then, the PLGA/TMP NDM began to decompose at 230 °C, while the PLGA NDM and PLGA/CS NDM did so at about 260 °C. The decomposition of the PLGA/CS NDM and PLGA/TMP NDM was completed at around 340 °C, while that of the PLGA NDM was done at about 360 °C. The ash residue of the NDM was around 2%. This shows that the CS and TMP had a weak influence on the thermal stability of the PLGA. However, the physiological temperature of the human body cannot interfere at all with the thermal stability of the systems investigated by thermogravimetry.

### 3.2. Encapsulation Efficiency and In Vitro TMP Release Profiles

According to our calculations, the EE (%) of TMP is (57.95 ± 2.46) %, and the working concentration of TMP can be reached after release (the standard curve used to obtain the value of the EE is plotted in the Appendix A). Figure 3 shows the in vitro release profile of the TMP (the associated calibration curve is displayed in the Appendix A). During the first 8 h, the TMP exhibited a burst release of more than 50%, reaching the working concentration that could play a neuroprotective role in the early stage. After this steep increase, the release of the TMP occurred. It was released moderately and sustainedly until the 14th day, when the process leveled off and the cumulative release ratio reached about 80%.

### 3.3. PLGA/CS NDMs Inhibit the Excessive Proliferation of Fibroblasts

During wound healing, fibroblasts are activated. Activated fibroblasts have higher cytoskeleton tension, indicated by obvious stress fibers and contractile phenotype, which enhance the secretion of ECM and promote wound healing and tissue regeneration. A balanced secretion of ECM is essential for wound healing, as the accumulation of excessive ECM can lead to the development of tissue adhesions and cause tissue fibrosis and scar tissue formation [44]. It has been reported that PLGA can reduce the formation of epidural fibrosis [45]. To investigate the effect of the PLGA NDM and PLGA/CS NDM on fibroblasts, we seeded fibroblasts on NDMs for culture and performed cell viability assays on the fibroblasts after 1, 3, and 5 days of cell culture. As shown in Figure 4A, cell viability was increased from day 1 to day 5, regardless of the NDM used for culture. However, cell viability was decreased in the PLGA NDM and PLGA/CS NDM compared to the control group (TCP), in which the cells were cultured under normal conditions, and the PLGA/CS NDM induced lower cell viability than that observed in the PLGA NDM, although there were no statistical differences between the two groups. This is consistent with the previous reports that CS can progressively inhibit the proliferation of fibroblasts in a CS dose-dependent manner [17].

From a morphological point of view, the fibroblasts of the TCP stained with FITC-Phalloidin were compared to the fibroblasts similarly stained and cultured in the presence of NDMs. Fibroblasts on the TCP group showed a normal long spindle shape, while fibroblasts on the PLGA NDM and PLGA/CS NDM groups became more elongated, grew more branched, and overall became irregular in morphology (Figure 4B–D), which means that both the PLGA NDM and PLGA/CS NDM could have affected the growth and morphology of the fibroblasts. These results are in agreement with those of the CCK8.

To further determine whether NDMs reduced fibroblast cell viability by decreasing cell proliferation, the fibroblasts were labeled as Ki-67, and the cell proliferation capacity was analyzed. The percentage of Ki-67-positive cells was significantly decreased in the PLGA NDM and PLGA/CS NDM groups compared to the TCP group, and this phenomenon was more pronounced in the PLGA/CS NDM group (Figure 5 and Appendix A), which was consistent with the previous CCK8 results. This suggests that the PLGA/CS NDM repressed cell viability by inhibiting the excessive proliferation of fibroblasts, which is in line with a previous report [46]. 

The cell viability of the fibroblasts showed an increasing trend in the PLGA/CS NDM group with the passage of culture time, meaning that the PLGA/CS NDM was capable of supporting tissue regeneration in our culture system. However, the PLGA/CS NDM was able to inhibit the excessive proliferation of fibroblasts and, thus, maintain a balanced secretion of ECM. During wound healing, an excessive proliferation of fibroblasts and excessive secretion of collagen type I can cause wound adhesion and ultimately lead to the formation of scar tissue. It has been reported that CS can inhibit the secretion of collagen type I with fibroblasts in scar tissue but has no effect on normal fibroblasts, which also prevents the formation of scar tissue [44,47]. In a few words, these results suggest that PLGA/CS NDM has the potential to both support wound healing and prevent dural adhesion and scar tissue formation. 

### 3.4. PLGA/CS NDMs Promote the Survival of OGD-Treated SH-SY5Y Cells

Previous studies have shown that CS degradation products (COS) can promote nerve repair by improving the local microenvironment (in particular, by stimulating Schwann cell proliferation), regulating macrophage migration, and alleviating the cell apoptosis of cortical neurons treated with glucose deprivation [21,48]. In order to investigate whether PLGA/CS NDMs have neuroprotective effects on ischemia–reperfusion brain injury, SH-SY5Y cells underwent an OGD treatment, and the immersion solution of the PLGA NDM and PLGA/CS NDM were mixed 1:1 with fresh complete medium for reperfusion to avoid the direct influence of the topological structure of the NDMs on SH-SY5Y cells. Cell viability was detected with CCK8 after 24 h of reperfusion. In Figure 6A, there was no significant difference in the cell viability between the PLGA NDM group and the OGD/R group, while the cell viability of the PLGA/CS NDM group was significantly improved. This result suggested that PLGA/CS NDM can promote the survival of SH-SY5Y cells treated with OGD/R. In view of these results, we believe that PLGA/CS NDMs are more suitable for neuroprotection as a sustained release carrier. 

### 3.5. The Working Concentration of TMP with Neuroprotective Effect

It has been reported that TMP shows good neuroprotective effects on OGD/R-treated SH-SY5Y cells and neurons in in vitro experiments. However, the working concentration of TMP is not identical in different reported investigations, which may be related to the state of the cells, the batch of the drug, the laboratory environment, or the treatment procedure [49,50,51]. To determine the working concentration of TMP upon OGD/R challenge, we used OGD-treated SH-SY5Y cells that were then reperfused with complete medium with different concentrations of TMP. After 24 h of reperfusion, the cell viability was detected with CCK8. The results show that the OGD/R treatment significantly reduced the cell viability of SH-SY5Y cells compared to the control group. TMP (50 μM) did not have an obvious protective effect against OGD/R-induced injury, while 100 μM, 200 μM, and 400 μM of TMP significantly improved the cell viability of OGD/R-treated SH-SY5Y cells, the last two concentrations having almost the same effect (Figure 6B). Therefore, we concluded that TMP has a neuroprotective effect on our culture system when the concentration in the culture medium exceeds 100 μM.

### 3.6. PLGA/TMP/CS NDMs Promote the Survival of OGD-Treated SH-SY5Y Cells

TMP has been shown to have significant neuroprotective effects on ischemia–reperfusion-induced brain injury and has been applied in clinical and basic research. In addition, TMP can diminish the proliferation of fibroblasts [52]. However, the clinical application of TMP has been limited due to its poor water solubility, short half-life, and low concentration of distribution at the injury site. To address these drawbacks, we fabricated a PLGA/TMP NDM with the coaxial electrospinning technique, in which the PLGA serves as a slow-release carrier for the TMP. The TMP was released into the medium by the degradation of the PLGA. To enhance the neuroprotective effect of the NDM, we fabricated a PLGA/TMP/CS NDM by grafting CS onto the PLGA/TMP NDM. However, we did not observe that the PLGA/TMP/CS NDM further reduced the proliferation of fibroblasts compared to the PLGA/CS NDM (Appendix A), possibly because the TMP concentration released from the NDM did not reach the working concentration of 400 μM to diminish the proliferation of fibroblasts, according to the report. To investigate the neuroprotective effects of the PLGA/TMP/CS NDM on ischemia–reperfusion-induced brain injury, SH-SY5Y cells were subjected to OGD treatment. We extracted the immersion solution of the PLGA/TMP/CS NDM after 1, 4, 7, and 14 days, and mixed the immersion solution 1:1 with fresh complete medium for reperfusion after the OGD treatment. After 24 h of reperfusion, the cell viability was tested with CCK8, and the LDH release was assayed. The CCK8 results show that the PLGA/CS NDM promoted the survival of SH-SY5Y as before. The cell viability was further increased with the 1-, 4-, 7-, and 14-day immersion solution of the PLGA/TMP/CS NDM groups compared to the PLGA/CS NDM group. From day 1 to day 14, the cell viability continued to increase (Figure 6C), which may have been due to the COS produced with the CS degradation and the TMP released from the NDM. However, the LDH release assay revealed that the effects of the NDM on LDH release were not obvious. Only 7- and 14-day immersion solution of the PLGA/TMP/CS NDM could significantly reduce the release of the LDH compared to the OGD/R group. (Figure 6D). In conclusion, these results indicate that PLGA/TMP/CS NDM may have long-term neuroprotective effects for ischemia–reperfusion-induced brain injury.

### 3.7. PLGA/TMP/CS NDM Promote Nerve Repair

TMP not only has neuroprotective effects, but it also enhances neurogenesis. COS can facilitate nerve regeneration. To investigate whether PLGA/TMP/CS NDM can promote the regeneration of neural tissue subjected to ischemia–reperfusion injury, we used OGD-treated SH-SY5Y cells and reperfused them with the previous immersion solution for different durations, as described in Figure 6C. The cells were labeled as GAP43. GAP43 is an axonal membrane protein involved in neural outgrowth, synapse development formation, and neural cell regeneration. The expression of GAP43 means that OGD-treated SH-SY5Y cells are undergoing neural regeneration. The labeling results show that the OGD/R treatment slightly increased the expression of the GAP43 protein, but there was no statistical difference. Both the PLGA/CS NDM and PLGA/TMP/CS NDM further increased the expression of the GAP43 protein, while the PLGA NDM did not have such an effect (Figure 7 and Appendix A). The results suggest that COS and TMP may promote the expression of GAP43 during NDM degradation. Regarding the effect of OGD/R treatment on GAP43 protein expression, the results of different studies were inconsistent. This may be related to factors such as the cell status or treatment process. Some researchers consider that OGD injury can activate the endogenous mechanisms of neuroprotection and neuroplasticity, which may promote the expression of GAP43 [53]. We believe that when cells grow well or when the OGD treatment time is short, GAP43 protein expression may be up-regulated. On the contrary, when the cell growth status is poor or the OGD treatment is severe, the expression of the GAP43 protein may be down-regulated.

To further clarify the influence of the PLGA/TMP/CS NDM on nerve repair, we stained the actin cytoskeleton and observed the cell morphology. We found that approximately 35% of the SH-SY5Y cells showed pyknosis and lost neurites most likely caused by OGD/R injury. The PLGA NDM and PLGA/CS NDM did not improve the cell morphology. The immersion solution of the PLGA/TMP/CS NDM reduced the damage to the neurites, and the proportion of cells without neurites gradually decreased from day 1 to 14 of the immersion solution (Figure 8A,B). These results indicate that PLGA/TMP/CS NDM can protect the neurites of OGD/R-treated SH-SY5Y cells during degradation. As for the reason the PLGA/CS NDM had no effect, we suppose that the level of COS could not reach the working concentration due to the short immersion time of the NDM, and the time of the cell culture was not enough. We next measured the length of the remaining neurites. As shown in Figure 8C, the lengths of the remaining neurites of the OGD-treated cells reperfused with the PLGA NDM immersion solution were significantly reduced, while they were significantly recovered by those reperfused with the PLGA/TMP/CS NDM immersion solution. We also noted that the PLGA/CS NDM seemed to have had a good effect on the recovery of the lengths of the neurites. Therefore, PLGA/TMP/CS NDM may promote nerve repair with COS and TMP.

Interestingly, we found that the results of the nerve repair did not exactly match the CCK8 results. According to our statistical results, the morphology of the OGD/R-treated SH-SY5Y cells seemed to be almost completely restored with the PLGA/TMP/CS NDM, but the cell viability was not regained to normal levels, which might have been a consequence of an insufficient analysis of the cell morphology. Furthermore, no in vivo experiments were carried out, so the physiological significance of PLGA/TMP/CS NDM needs to be further studied. In conclusion, our results suggest that PLGA/TMP/CS NDM systems inhibit the excessive proliferation of fibroblasts in vitro and promote the survival and neural repair of OGD/R-treated SH-SY5Y cells in our culture system. Thus, PLGA/TMP/CS NDM may be an alternative for dural grafts.

## 4. Conclusions

In this paper, we report some PLGA/TMP NDM structures with antifibrotic and neuroprotective effects fabricated with the coaxial electrospinning technique, where the PLGA component was chosen as a slower release carrier. To neutralize the acidic environment generated by PLGA degradation and to enhance the beneficial effect of the NDM (antifibrotic, neuroprotective), we prepared PLGA/TMP/CS NDM by grafting CS on the NDM surface via the EDC/NHS cross-linking method. We found that all NDMs inhibited the excessive proliferation of fibroblasts, but the PLGA/TMP/CS NDM systems were more effective. In terms of neuroprotection, the PLGA/TMP/CS NDM structures were able to promote the survival of OGD/R-treated SH-SY5Y cells as well as nerve repair. In conclusion, the study suggests that PLGA/TMP/CS NDMs may have long-lasting neuroprotective effects and prevent tissue adhesion, fibrosis, and scar tissue formation as dural grafts in undesirable cases of dural injuries.

## Figures and Tables

**Figure 1 polymers-14-01882-f001:**
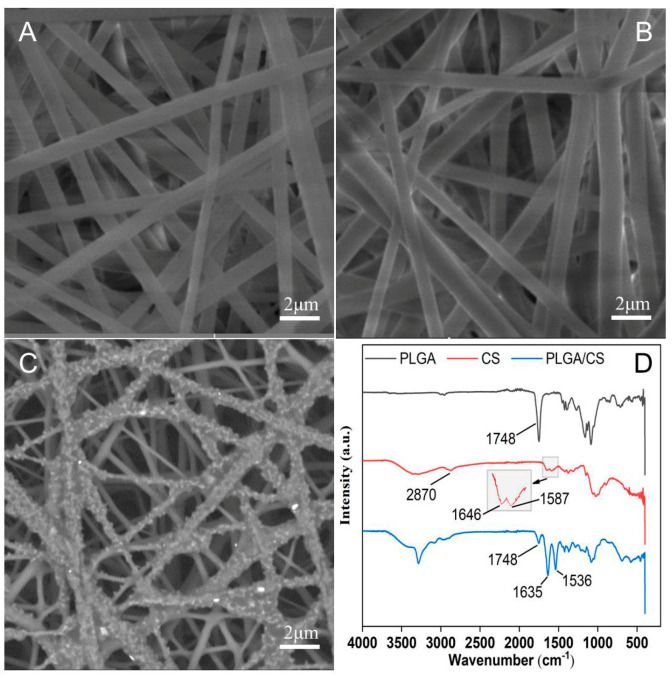
(**A**–**C**) SEM images showing the PLGA NDM (**A**), the PLGA/TMP NDM (**B**), and the PLGA/CS NDM (**C**). (**D**) FTIR spectra of the PLGA NDM, CS, and PLGA/CS NDM.

**Figure 2 polymers-14-01882-f002:**
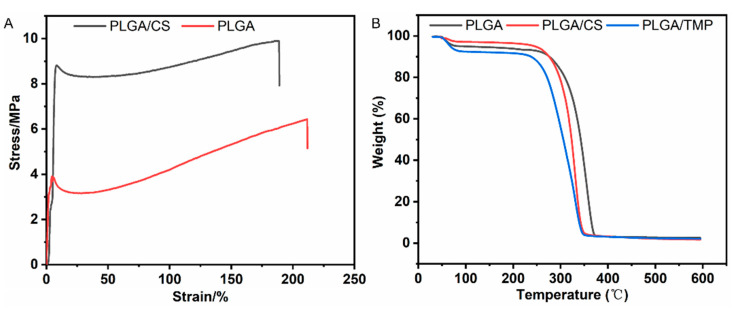
(**A**) Stress–strain curves of PLGA NDM and PLGA/CS NDM. (**B**) TGA profiles of PLGA NDM, PLGA/CS NDM, and PLGA/TMP NDM.

**Figure 3 polymers-14-01882-f003:**
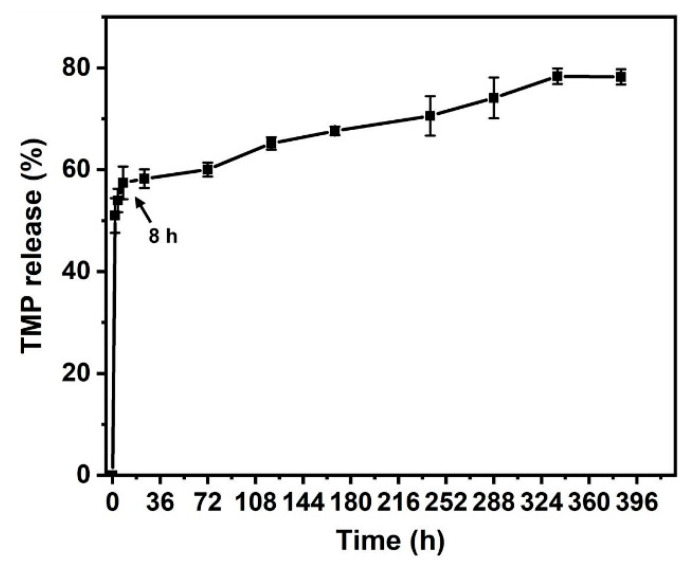
TMP release profiles in vitro from PLGA/TMP/CS NDM (measurements were performed in triplicate). During the first 8 h, the TMP exhibited a burst release of more than 50%.

**Figure 4 polymers-14-01882-f004:**
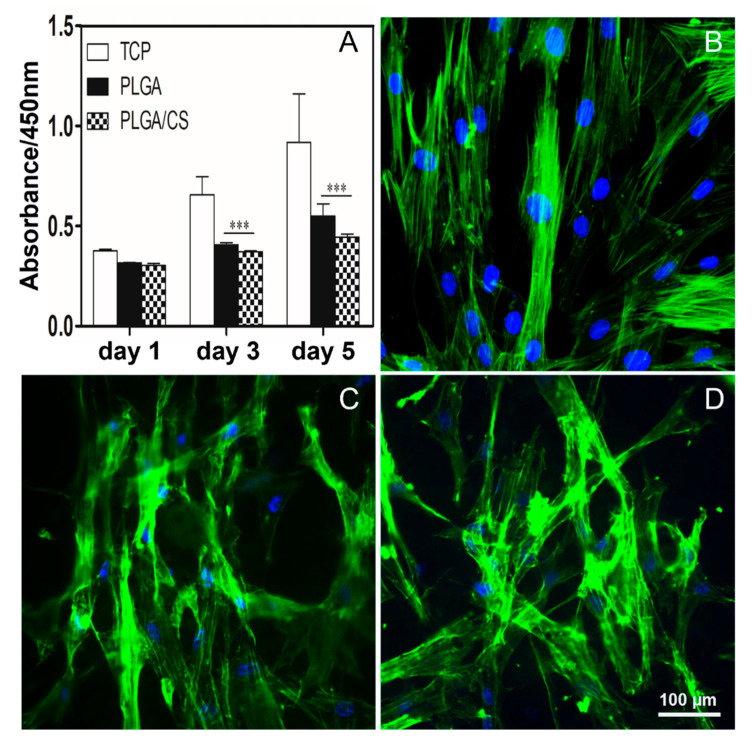
(**A**) Cell viability of fibroblasts seeded on glass slides as the control group (TCP), on PLGA NDM and PLGA/CS NDM after culture of 1, 3, and 5 days. *** *p* < 0.001 as compared with TCP. (**B**–**D**) Fluorescence micrographs showing the morphology of the fibroblasts after 5 days of culture. (**B**) TCP cells. (**C**) Cells seeded on PLGA NDM. (**D**) Cells seeded on PLGA/CS NDM. The cell nuclei were stained with DAPI (blue), and the actin cytoskeleton was stained with Phalloidin-iFluor 488 (FITC, green). Data of 3 replicates are plotted in Figure 4A.

**Figure 5 polymers-14-01882-f005:**
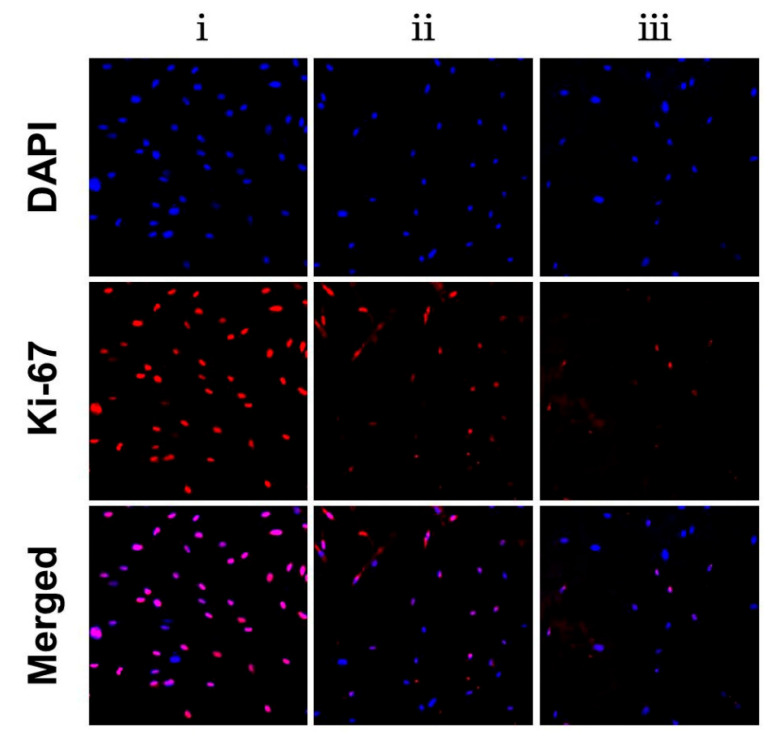
Fluorescence micrographs displaying Ki-67-positive fibroblasts seeded on TCP (**i**), PLGA NDM (**ii**), and PLGA/CS NDM (**iii**) after 5 days of culture. The cell nuclei were stained with DAPI (blue) and the cell nuclei with Ki-67-positive were labeled with red, meaning the cells were proliferating.

**Figure 6 polymers-14-01882-f006:**
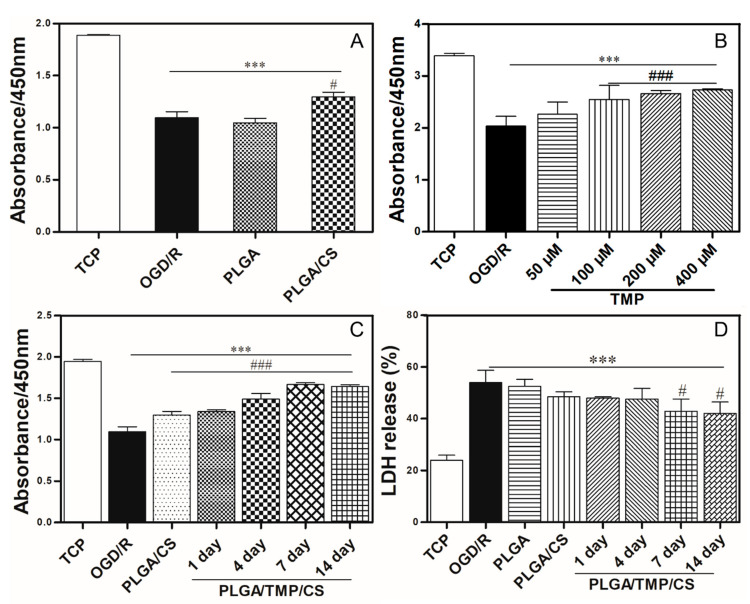
(**A**) Cell viability of OGD-treated SH-SY5Y cells—treated for 3 h and reperfused with immersion solution of PLGA NDM and PLGA/CS NDM mixed 1:1 with fresh complete medium. (**B**) Cell viability of OGD-treated SH-SY5Y cells—treated for 3 h and reperfused with complete medium containing different concentrations of TMP. *** *p* < 0.001 as compared with the TCP. ^#^
*p* < 0.05 and ^###^ *p* < 0.001 as compared with the cells reperfused only with complete medium. (**C**) Cell viability of OGD-treated SH-SY5Y cells—treated for 3 h and reperfused with immersion solution of PLGA/CS NDM and PLGA/TMP/CS NDM. The durations of the PLGA/TMP/CS NDM immersion were 1 day, 3 days, and 5 days, respectively, to obtain the corresponding immersion solutions. (**D**) LDH release assay of OGD-treated SH-SY5Y cells—treated for 3 h and reperfused with immersion solution of PLGA NDM, PLGA/CS NDM, and PLGA/TMP/CS NDM.

**Figure 7 polymers-14-01882-f007:**
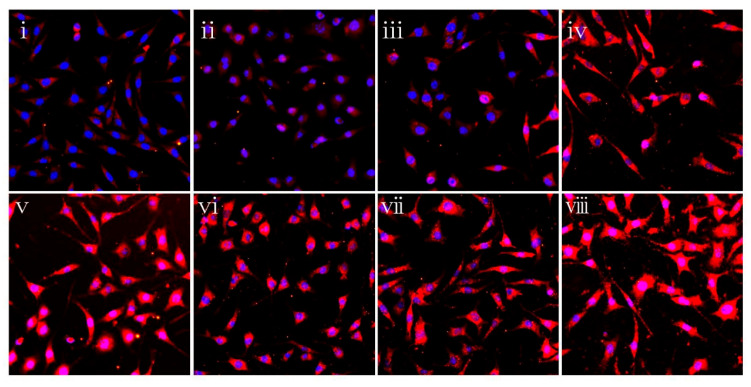
Fluorescence micrographs showing the expression of GAP43 of SH-SY5Y cells. (**i**) TCP cells. (**ii**) The cells treated with OGD for 3 h and reperfused only with complete medium. (**iii**,**iv**) The OGD-treated cells reperfused with immersion solutions of PLGA NDM (**iii**) and PLGA/CS NDM (**iv**). (**v**–**viii**) The OGD-treated cells reperfused with immersion solution of PLGA/TMP/CS NDM, with durations of NDM immersion of 1 day (**v**), 4 days (**vi**), 7 days (**vii**), and 14 days (**viii**).

**Figure 8 polymers-14-01882-f008:**
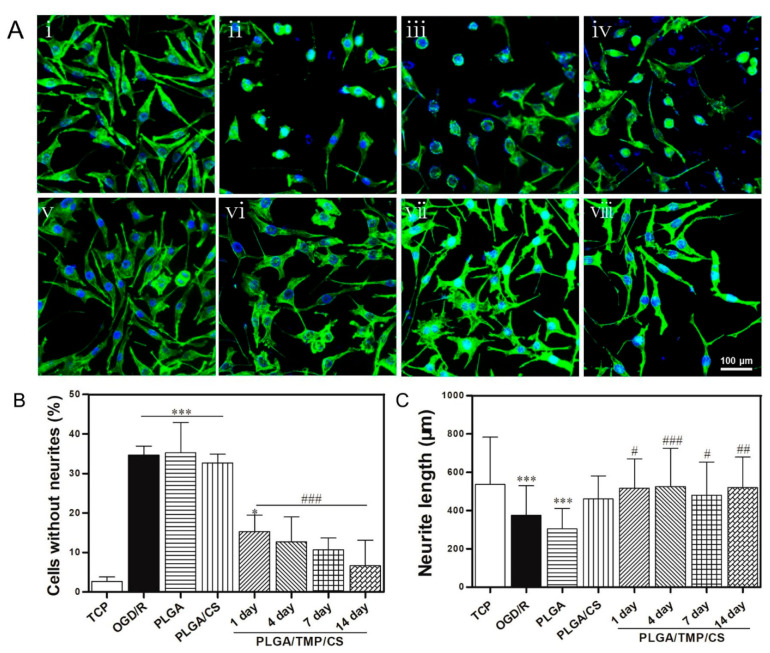
(**A**) Fluorescence micrographs showing the morphology of SH-SY5YS cells treated with OGD/R for 2 days. (**i**) TCP cells. (**ii**) The cells treated with OGD for 3 h and reperfused only with complete medium. (**iii**,**iv**) The cells treated with OGD and reperfused with immersion solution of PLGA NDM (**iii**) and PLGA/CS NDM (**iv**). (**v**–**viii**) The cells treated with OGD and reperfused with immersion solution of PLGA/TMP/CS NDM, the durations of which were 1 day (**v**), 4 days (**vi**), 7 days (**vii**), and 14 days (**viii**). (**B**) The percentage of cells after 2 days of OGD/R treatment. (**C**) Lengths of neurites of the cells after 2 days of OGD/R treatment. * *p* < 0.05 and *** *p* < 0.001 as compared to TCP. ^#^ *p* < 0.05, ^##^ *p* < 0.01 and ^###^
*p* < 0.001 as compared to OGD/R.

## Data Availability

The original contributions presented in the study are included in the article/Appendix A; further inquiries can be directed to the corresponding author.

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
