# Peer review of "Design and Fabrication of Nanofibrous Dura Mater with Antifibrosis and Neuroprotection Effects on SH-SY5Y Cells"

_polymers, 2022, doi:10.3390/polym14091882_

Round 1
Reviewer 1 Report
Suggestions/comments regarding the manuscript are inserted into the attachment.

Reviewer 2 Report
In this manuscript, PLGA, TMP, and chitosan were used to prepare a nanofibrous dura mater that can play a neuroprotective role to prevent the formation of dura adhesion and scar tissue. The present form of the manuscript would be acceptable for publication after minor revision:
- The mechanical performance of the fabricated nanofibrous dura mater is relevant to report. In fact, the biological activities of cells are significantly influenced by the biomechanical features of an implant. We suggest investigating the effect of chitosan on the elasticity \stiffness of the fibers.
- DSC, TGA, and DMTA could be performed to determine the effects of TMP and chitosan on the mechanical and physicochemical features of the PLGA.
Round 2
Reviewer 1 Report
Suggestions/comments regarding the manuscript are inserted into the attachment.
